# Vesicovaginal Fistulas: Prevalence, Impact, and Management Challenges

**DOI:** 10.3390/medicina59111947

**Published:** 2023-11-03

**Authors:** Orawee Chinthakanan, Pokket Sirisreetreerux, Apisith Saraluck

**Affiliations:** 1Female Pelvic Medicine and Reconstructive Surgery Division, Department of Obstetrics & Gynecology, Ramathibodi Hospital, Mahidol University, Bangkok 10400, Thailand; 2Urology Division, Department of Surgery, Ramathibodi Hospital, Mahidol University, Bangkok 10400, Thailand; pokket.sir@mahidol.ac.th

**Keywords:** fistula correction, genital fistula, management, prevalence, quality of life, vesicovaginal fistula

## Abstract

*Background and Objectives:* Vesicovaginal fistulas (VVFs) are an abnormal communication between the vagina and bladder and the most common type of acquired genital fistulas. This review will address the prevalence, impact, and management challenges of VVFs. *Materials and Methods:* Epidemiologic studies examining VVFs are considered. In addition, publications addressing the treatment of VVFs are reviewed. *Results:* VVFs in developing countries are often caused by obstructed labor, while most VVFs in developed countries have iatrogenic causes, such as hysterectomy, radiation therapy, and infection. The reported prevalence of VVFs is approximately 1 in 1000 post-hysterectomy patients and 1 in 1000 deliveries. VVFs affect every aspect of quality of life, including physical, mental, social, and sexual aspects. Prevention of VVFs is essential. Early diagnosis is necessary to reduce morbidity. Nutrition, infection control, and malignancy detection are important considerations during evaluation and treatment. Conservative and surgical treatment options are available; however, these approaches should be customized to the individual patient. The success rate of combined conservative and surgical treatments exceeds 90%. *Conclusions:* VVFs are considered debilitating and devastating. However, they are preventable and treatable; key factors include the avoidance of prolonged labor, careful performance of gynecologic surgery, and early detection.

## 1. Introduction and Prevalence

Urogynecologic fistulas include vesicovaginal, vesicouterine, ureterovaginal, urethrovaginal, and combined fistulas (Figure 1). A vesicovaginal fistula (VVF) refers to an abnormal connection between the urinary bladder and the vagina, leading to persistent urinary leakage characterized by the spontaneous discharge of urine through the vaginal canal [1]. A vesicovaginal fistula (VVF) represents the most common type of acquired fistula. Different parts of the world have various common causes, etiologies, and management trends, which is challenging for healthcare professionals. The most prevalent cause of bladder injury in North America is bladder injury during a hysterectomy [2]. In Western nations, gynecological surgery is the most prevalent cause of vesicovaginal fistulas [3]. According to numerous countries, obstructed labor resulting in pressure necrosis is the most prevalent cause of vesicovaginal fistulas [4]. As most vesicovaginal fistulas require definitive surgical treatment, a thorough diagnostic evaluation is required. Timing, strategy, graft utilization, postoperative care, and surgical skill must be optimized for successful management. Due to the fact that VVF prevalence, etiology, risk factors, development, and disease-specific management techniques have been reported in a variety of ways, the purpose of this review is to identify and summarize the most important challenges in all aspects of VVFs.

The classification of VVFs is mostly defined based on size in the majority of the literature. Typically, simple fistulas manifest as isolated nonradiated fistulas that are rather small, measuring ≤0.5 cm in size. Complex fistulas include cases where previous attempts at fistula repairs have been unsuccessful or involve fistulas that are larger in size (≥2.5 cm). These types of fistulas are often associated with chronic conditions or radiation. The majority of publications in the field of medical research classify fistulas with sizes ranging from 0.5 to 2.5 cm as complicated in nature [5]. Nevertheless, several academic studies have provided definitions of VVFs based on their anatomical position, including trigonal, supratrigonal, and other descriptions such as circumferential, juxtacervical, and juxtaurethral [1].

The first record of a VVF was found in the Ebers Papyrus from Egypt in 1550 BC [6]. The oldest evidence of a VVF was found by Professor Derry in Cairo in the mummy of Queen Henhenit, who reigned around 2050 BC [7]. In 1663, the first surgical correction technique was published in *Operative Gynaecology* by Hendrik van Roonhuyse. He proposed the theory of VVF treatment as (1) proper exposure of the fistula with a speculum, (2) denudation exclusive of the bladder wall, and (3) approximation of the denuded edges using a stitching needle made of a stiff swan quill [8]. Johann Fatio reported the first successful surgical repair of two VVFs using the van Roonhuyse technique in 1675 [9]. Levret was the first to introduce the knee–chest position to facilitate VVF repair [6]. James Marion Sims, known as the “father of modern gynecology,” dedicated his work to the treatment of VVFs and established the first women’s hospital in the United States [10]. He adopted a surgical technique using vaginal skin flaps and tension-free closure of the VVF. He reported successful VVF repair using silver wire after 30 attempts on his slave in 1852 [11]. His published article became a landmark paper [12] for gynecologists and sparked controversy in the field of medical ethics [13,14]. There have been many publications discussing alterations in surgical techniques for the development of vesicovaginal fistula (VVF) repair in that era. For instance, Collis (1861) reported on the use of multilayered suturing for VVF closure, while Schuchardt (1893) described a technique including incision to access the pararectal area, with the goal of enhancing exposure. In the 1880s, Maisonneuve documented a technique that included the separation of the bladder from the vagina for the purpose of repairing VVFs [15]. In 1888, the first successful transabdominal VVF repair was reported by Trendelenburg [16]. This procedure pioneered the abdominal suprapubic approach. Later, the bivalve supratrigonal repair technique called the O’Conor technique was introduced in 1950 and became a standard technique for transabdominal repair [17]. The conventional O’Connor technique uses suprapubic access for extraperitoneal dissection of the retropubic space to dissect the urinary bladder, followed by a long sagittal cystotomy (bivalving the bladder) conducted to the fistula [18]. The fistulous tract is then excised, and two-layered closure is performed after tissue transposition between the vaginal and bladder walls. This technique is widely recognized as one of the most renowned methods for repairing VVFs and continues to be utilized in contemporary practice. It has undergone adaptations and has been documented as a modified version of the O’Connor technique. Latzko introduced a vaginal technique in 1942 [19]. The Latzko technique was considered the standard vaginal repair technique [11]. However, vaginal shortening is a consequence of the vaginal approach. From the foundations of VVF repair techniques to the current standard of surgical treatment, the evolution of treatment techniques has correlated with the progression of time.

Because of their high morbidity, VVFs are a public health concern as determined by the global burden of disease from the World Health Organization. Although VVFs are uncommon in developed countries, they are a common complication of childbirth resulting from prolonged obstructed labor in developing countries [20]. A recent published systematic review and meta-analysis reported that the pooled prevalence of obstetric fistulas was 0.29 fistulas per 1000 reproductive-aged women and that the pooled incidence was 0.09 per 1000 recently pregnant women [21]. Estimates suggest that at least 3 million women in poor countries have an unrepaired VVF and that 30,000 to 130,000 new cases develop each year in Africa alone [20]. The general public and the world medical community remain largely unaware of this problem. Fistulas caused by obstructed labor were eradicated from industrialized nations by the middle of the 20th century as effective systems of obstetric care were developed to cover the entire population of childbearing women [20]. Obstetric fistulas remain a public health problem in developing countries within Sub-Saharan Africa and South Asia, where maternal mortality rates are nearly 100 times higher than rates in developed countries [22]. The prevalence of VVFs after a hysterectomy is 0.8 to 1.0 per 1000 women [23,24].

## 2. Etiology

Risk factors for obstetric fistulas historically included primiparity [25,26], small and short body habitus (44 kg and <150 cm), and early marriage (15.5 years of age); however, they now include divorce or separation, lack of education, poor economic conditions, and residence in a rural area [26]. In addition, multiparous women and those with a history of Caesarean section are four times more likely to develop a high tract fistula [27]. Important risk factors for VVFs following benign gynecologic surgery include difficult cases, long operative times, large fibroids, and obese patients [24]. Excessive blood loss and current smoking may also create an environment suitable for VVF formation because of a delayed healing process and decreased tissue perfusion [28]. VVFs have several potential etiologies. Foreign bodies can cause VVFs (e.g., tampons, pessaries, and similar items) [29,30,31,32,33]. Such objects can cause tissue ischemia and subsequent tissue necrosis secondary to external pressure (crush/clamp injury), kinking of urinary tract tissue (when in close proximity to a ligated pedicle), or marked inflammation. Alternatively, direct laceration or a puncture injury to the urinary tract can result in immediate urinary leakage. Delayed injury may occur from retroperitoneal fibrosis, tissue pressure, or partial obstruction. A VVF may also develop during the healing process after pelvic surgery or radiation. Pelvic radiotherapy for malignancy decreases the blood supply, leading to tissue necrosis, sloughing, and fistula formation, even after many years [34]. Intriguingly, due to the fact that the incidence of placenta accreta spectrum is increasing as a result of the rising Cesarean section rate, some research suggests that this may be a future risk factor for an increase in VVFs [35].

## 3. Impact

Obstetric fistulas are associated with maternal mortality. According to the World Health Organization’s Global Burden of Disease study, 21.9% of the disability-adjusted life years lost by women aged 15 to 44 years were attributable to reproductive-related health, and 14.5 years per woman were lost to adverse maternity-related causes. Obstructed labor, which is a primary cause of maternal death, accounted for 22% of all morbid maternal conditions. Fistula formation caused by obstructed labor was eradicated from industrialized nations by the middle of the 20th century as effective systems of obstetric care were developed to cover the entire population of childbearing women [20]. However, obstetric fistulas still occur and are a major problem in developing countries. VVFs not only affect physical health but also have social, economic, emotional, and psychological consequences. In one meta-analysis, 36% of women with fistulas were divorced or separated [36]. In addition, 85% of women with fistulas incurred fetal loss from the delivery, which had a psychological impact [36]. Successful VVF repair significantly improves quality of life, including physical health, mental health, self-esteem, and social reintegration [36,37]. The latest study in 2023 found that there was a significant correlation between health stigma and psychological distress among VVF patients. The findings highlighted the importance of health disclosure in the management of the stigma’s effects on VVF patients’ health and well-being [38].

## 4. Management Challenges

Early detection and prompt treatment are essential for VVF management. Most patients complain of continuous urinary leakage about 7 to 10 days after gynecologic surgery. However, the onset of VVFs varies from immediately postoperatively to 6 weeks postoperatively. VVFs secondary to radiation therapy can take years to develop. No standard guidelines have been established in VVF management algorithms. Existing evidence is inconsistent, and no definitive recommendations can be made. According to most studies, surgical treatment is preferred to conservative treatment such as prolonged catheter drainage, fulguration, and glue/fibrin injections. The success rate depends on many factors such as the size, location, and cause of the VVFs. Surgical treatment can be performed with vaginal, abdominal, laparoscopic, robotic, and transurethral approaches.

## 5. Diagnosis

The patient’s medical, obstetric, and surgical history as well as a carefully performed pelvic examination are important for the diagnosis of a VVF. Confirmation of urine leakage is crucial to identify the site and size of a VVF and can be performed by a speculum examination. The physician can also evaluate the tissue quality prior to correction via a speculum examination. A biopsy is recommended in patients with a history of cancer. The double-dye technique can be used to differentiate between a ureterovaginal fistula and a VVF. Cystoscopy and voiding cystourethrography are useful to determine the size, site, number, and location of VVFs. Imaging studies (computed tomography and magnetic resonance imaging) are advantageous for the diagnosis of ureterovaginal fistulas, 12% of which are concurrent with a VVF (Figure 2).

## 6. Treatment of VVFs

### 6.1. Conservative Management

Small, uncomplicated VVFs measuring <1 cm are often initially managed with conservative treatment. Prolonged catheter drainage for 12 weeks is usually the initial management technique. Several other conservative treatment options are available, including transvaginal injection of fibrin sealant [40,41,42], Nd:YAG laser welding [43,44], cystoscopic electrocoagulation/fulguration/catheterization [45,46], endovaginal application of cyanoacrylic glue, application of platelet-rich plasma/fibrin-rich glue [47], curettage of the fistula tract using an ordinary metal screw [48], and use of the ball counter-pressure technique with a rubber or metal ball [49]. The success rate varies from 67% to 100%. Anticholinergic medication should be administered during conservative management.

In VVFs caused by the long-term use of vaginal pessaries, there were around 12 cases reported in the literature [50,51,52]. Most cases had to receive surgical repair. There was a case reported from Korea after a wait of 3–6 months following an injury with the application of topical estrogen and an indwelling catheter. They reported the first successful case of conservative treatment without surgical repair [53]. 

It has been reported that fibrin sealant and collagen as supplementary devices have been used successfully to treat VVFs. After electrocoagulation and catheter drainage for several weeks, this substance may be placed in the fistulous tract. This gel-like substance obstructs the tract until new tissue develops from its margins. However, the number of studies reported was small and there was a failure incidence [54]

Several studies have indicated that conservative management may be a viable treatment option for uncomplicated vesicouterine fistulas with a diameter of less than 0.5 cm. In the event that the illness does not exhibit resolution within a span of two months, it is advisable to contemplate surgical intervention as a means of mitigating the risk of additional complications [55].

### 6.2. Surgical Management

Many factors should be addressed before surgical repair of VVFs, including the timing of surgery, the surgical approach, whether to excise the fistulous tract, and the use of an interposition graft/flap. The Waaldijk classification of VVFs [56] classifies VVFs into types I, IIa, and IIb. This classification can be beneficial for selecting the optimal surgical approach. Surgical approaches include the transvaginal, transabdominal, and transvesical approaches. Surgical treatment can also be performed with a combination of approaches. The first surgical repair of a VVF is the most important predictive factor for successful treatment. The route and technique of the repair depend on the surgeon’s preference; the surgeon will perform the repair he or she feels the most experienced and comfortable executing. The most important criteria for effective surgical repair include good visualization of the VVF, thorough dissection of the fistulous tract, good approximation of the surgical margin, tension-free watertight suturing, use of a vascularized tissue interposition flap, overlapping of the suture line, and adequate bladder drainage. Most successful surgical repairs of VVFs involve the transvaginal approach. The most common surgical technique is Latzko’s colpocleisis, which is easily performed by dissecting the fistulous tract and surrounding tissue by about 2 to 3 cm. The defect is then closed with multiple layers of absorbable sutures (Figure 3). Tissue interposition using a peritoneal, gracilis muscle, omental, peritoneal, Martius labial fat flap, or a bladder mucosa autograft can be performed to increase the vascular supply and thus increase the success rate.

A transabdominal approach, including open, laparoscopic, and robotic-assisted laparoscopic techniques, is the second most common approach to VVF surgical treatment. This approach is recommended if the VVF is high or complex, involves the trigone area, requires ureteral reimplantation, is associated with a low bladder capacity that requires bladder augmentation, or is difficult to approach through the vagina. The laparoscopic and robotic techniques provide similar success rates. Omental flap interposition does not show a significantly higher success rate [17]. Although omental flap interposition is not routinely performed, it should be performed in patients with recurrent VVFs, large fistulas, irradiated tissue, or obstetric fistulas. The transabdominal approach may be performed using either a transvesical or extravesical technique. The traditionally performed technique is the transvesical bivalve bladder technique (O’Conor technique) (Figure 4a–e).

In surgical treatment, a hybrid technique involving direct transvesical insertion of 3 mm laparoscopic trocars and instruments guided by cystoscopy was reported to treat recurrent VVFs. This technique was selected as the least invasive and most ergonomic option described. Two years later, the patient was asymptomatic and there was no fistula recurrence [57].

### 6.3. Current Trends in Surgical Management and Surgical Technics

The systematic review of the literature on the use of laparoscopic and robotic-assisted VVF repair revealed 44 studies that included the robotic-assisted approach, laparoscopic single-site procedures, and conventional laparoscopic approaches. The results reveal an equivalent number of transvesical and extravesical procedures. With a follow-up period of 1 to 74 months, the success rate of laparoscopic VVF repair ranged from 80% to 100%. The transvesical and extravesical success rates were 95.89% and 98.04, respectively. In the hands of experienced surgeons, laparoscopic and robotic-assisted extravesical VVF repair is a safe, effective, and minimally invasive technique with outstanding cure rates comparable to those of the conventional transvesical approach [17].

A Consensus Report from the European Association of Urology Robotic Urology Section Scientific Working Group for Reconstructive Urology reported the best practices in the robotic-assisted repair of vesicovaginal fistulas and recommended preoperative fistula marking with a guidewire or ureteral catheter and placement of a protective JJ stent. An extravesical robotic method allows for extensive vesicovaginal dissection and bladder and vaginal mobilization with an excellent anatomic view. Fistula edges need careful sharp dissection. Tension-free bladder closure is crucial. Using tissue interposition works well. Series success rates are frequently around 100%. Postoperative bladder catheterization should last 10 days. In the deep pelvic region of patients with fragile anatomical features, robotic assistance allows the dissection of the vesicovaginal area, harvesting of a well-vascularized tissue flap, and tension-free bladder closure with minimum morbidity [58].

In 2023, “Vaginal-Laparoscopic Repair (VLR) of Primary and Persistent Vesico-Vaginal Fistula” was reported as safe and effective. This technique started with exposed VVFs and vaginal blades, then a Foley catheter to the bladder. Monopolar coagulation circumferentially delimited the vaginal VVF, leaving at least 1 cm of healthy tissue around the borders. Vaginal pressure with a gauze pad or bowel dilator in a glove revealed the vault. After thorough adhesiolysis and bladder–vaginal vault tissue dissection, a laparoscopic VVF was exposed. The distal catheter with channels was exposed by monopolarly opening the anterior vaginal wall. The anterior vaginal wall was longitudinally opened to reach the VVF after pulling the latter section in the abdomen. Full vaginal resection surrounding the catheter removed the catheter from the vaginal mucosa. The VVF’s vesical portion contained the Foley catheter at this point. A circumferential incision of the vesical region of the VVF around the catheter and thorough tissue resection were performed after enough room was available distally to implant the catheter in the bladder. Traction on the catheter helped stitch the bladder in two layers at the hole’s edges. After Foley deflation, the bladder was fully sutured. Deflating the balloon before tying the first stitch removed the Foley catheter. The vagina was sutured intracorporeally [59].

Another new technique was reported called “Natural Orifice Transurethral Endoscopic Vesicovaginal Fistula (NOTE-VVF)” for treatment of early and small fistulas. This technique used an optical lens and a 5 mm laparoscopic trocar inserted into the bladder through the female urethra. Subsequently, the VVF line was sutured at 1–2 mm intervals, and the bladder side was closed using 4/0 absorbable loop back v-loc sutures that were advanced into the bladder through the working trocar. In this case, VVF repair was performed with the NOTE-VVF treatment technique and reported a good outcome [60].

### 6.4. Timing of VVF Repair

There are no definitive data regarding the most appropriate timing for VVF repair. Both the timing and route of repair are best tailored to the individual patient [61]. The general consideration is the presence of healthy surrounding tissue. The healing of inflammation, infection, and tissue necrosis takes 2 to 3 months. Therefore, delayed VVF repair increases the chance of success [34]. However, delayed treatment of a VVF impacts the patient’s quality of life and has detrimental social consequences [36]. Most studies of VVFs did not state the optimal time range for repair. The timing of VVF repair can be categorized into early and late repair. Early repair is performed immediately after to 4 months after detection of the VVF. The most common cut-off is 6 weeks. Repair can be performed if the surgeon is able to identify the VVF at the time of gynecologic surgery or obstetric injury. The success rate does not significantly differ between early and late repair (86–100% and 67–100%, respectively) [61]. Experts generally recommend a 4- to 6-week interval from the onset of the fistula to surgical treatment [62]. This interval should be 2 to 3 weeks for the transvaginal approach and 3 to 6 months for the abdominal approach. With respect to the cause of the VVF, uncomplicated postgynecologic fistulas should be repaired immediately. The interval should be 3 to 6 months after obstructed labor and 6 to 12 months after radiotherapy.

### 6.5. Postoperative Care

An important issue of postoperative care is to ensure the maintenance of bladder emptying by preventing clot obstruction and retaining the urethral catheter. Bladder drainage is usually accomplished by an indwelling urethral catheter for 10 to 14 days for simple fistulas and 14 to 21 days for complex fistulas. Placement of a suprapubic cystostomy drainage tube is an option. Ambulation is encouraged, and appropriate prophylactic antibiotics are generally given. Anticholinergic medication is necessary to avoid bladder spasms. The closed drainage system is usually removed on postoperative day 2 or 3. A retrograde cystogram may be obtained before catheter removal to ensure fistula closure. Patients are advised to avoid the use of tampons and refrain from sexual activity for 2 months after the procedure.

### 6.6. Prevention

The estimated incidence of bladder injury associated with gynecologic surgery is 3.2% to 4.8% [63,64,65]. Good surgical techniques and early bladder injury detection are the keys to VVF prevention. Good surgical techniques to avoid bladder injury include the identification, dissection, and reflection of all contiguous lower urinary tract structures during gynecologic surgery. If bladder injury occurs despite these efforts, the surgeon should strive for intraoperative recognition and repair. Routine intraoperative cystoscopy after all major gynecologic operations may facilitate the recognition of a real or potential injury, allowing intraoperative repair [66]. Immediate repair is easier, more successful, and less morbid for the patient.

For obstetric fistulas, the quality of intrapartum care is crucial. Improved access to standard obstetric care, avoidance of prolonged labor, avoidance of unnecessary vaginal obstetric procedures, and drainage of the bladder during labor are useful preventive factors [20,67].

## 7. Novel Treatment of VVFs

Some interventions were reported as novel techniques for treating VVFs. Using an interposing layer of fibrin glue to reduce the failure rate of laparoscopic vesicovaginal fistula repair was inspired by the idea of applying an omental flap to reduce the failure rate of the procedure. Using fibrin glue in one group and an omental flap in another, forty patients with a relatively large vesicovaginal fistula are compared in a study with two arms. This result is statistically significant; therefore, laparoscopic repair of vesicovaginal fistulas should consider the use of fibrin glue [68].

Platelet-rich plasma (PRP) was utilized in a novel treatment study for VVFs. In 12 patients with iatrogenic VVFs, studies evaluated the effect of concurrent PRP injections into the tissue surrounding the fistulas and PRF glue inserted into the VVF tract. After three months, 91.67 percent of patients exhibited complete dryness post-injection and normal cystography and physical examinations. In another investigation, PRP was applied to the fistula of 16 patients between six and eight weeks prior to the Latzko procedure for VVF repair. One patient was cured by PRP injection without the need for surgery [69,70].

## 8. Conclusions

VVFs are considered debilitating and devastating. However, they are preventable and treatable; key factors include the avoidance of prolonged labor, careful performance of gynecologic surgery, and early detection. Improvement in obstetric care in developing countries is crucial. Treatment of VVFs with minimally invasive procedures will become the new standard. Further studies are necessary to develop new treatment modalities that improve women’s quality of life.

## Figures and Tables

**Figure 1 medicina-59-01947-f001:**
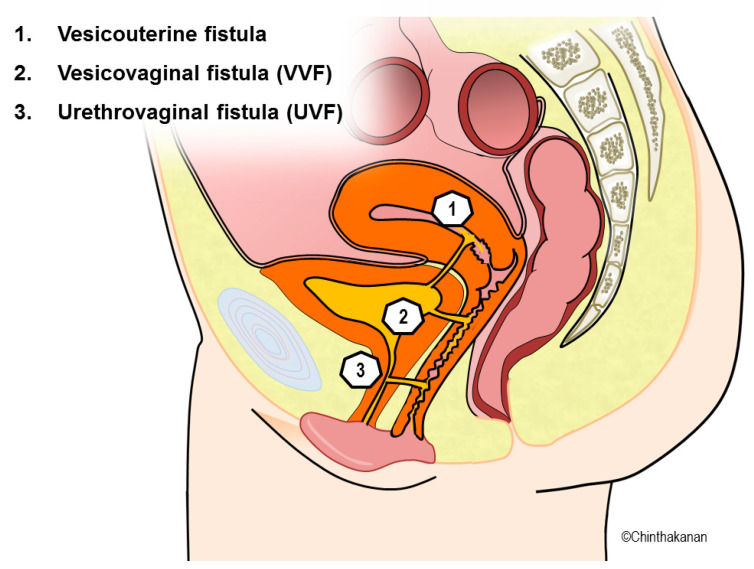
Urogynecologic fistulas.

**Figure 2 medicina-59-01947-f002:**
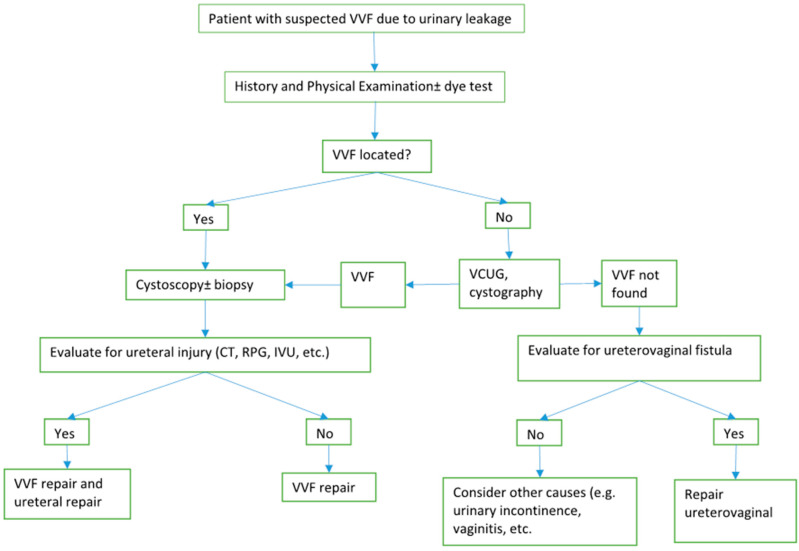
Flow chart of VVF management [39].

**Figure 3 medicina-59-01947-f003:**
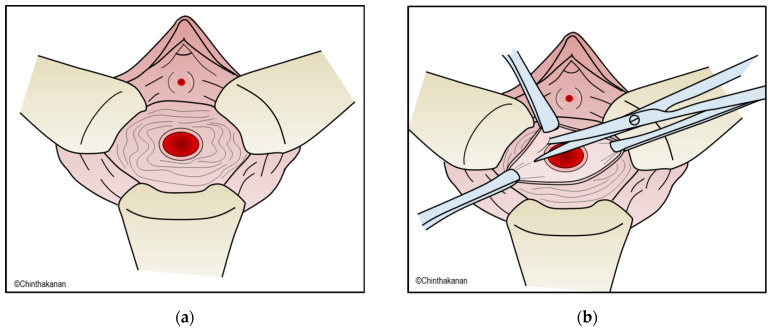
(**a**–**d**) Latzko’s colpocleisis.

**Figure 4 medicina-59-01947-f004:**
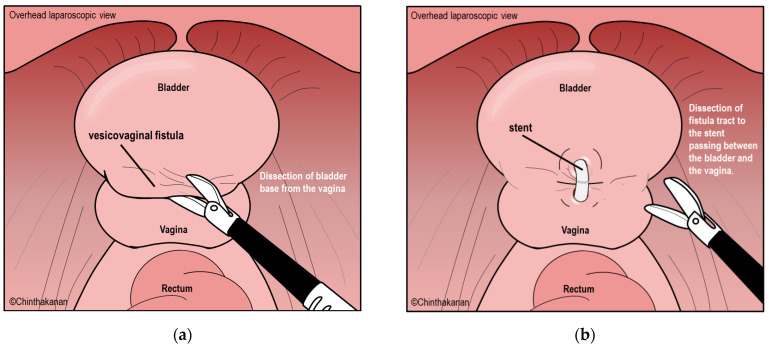
(**a**–**e**) Laparoscopic extravesical technique.

## Data Availability

There was no new data create. (Review article).

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
