# Peer review of "Vesicovaginal Fistulas: Prevalence, Impact, and Management Challenges"

_medicina, 2023, doi:10.3390/medicina59111947_

Round 1
Reviewer 1 Report
Comments and Suggestions for Authors
This review addresses the prevalence, impact, and management challenges of VVF.
The main concern is the lack of work novelty. The authors should emphasize the novelty of this study since there are many review articles on this topic.
Comments on the Quality of English Language
Minor editing of English language required.
Author Response
Dear Reviewer(s) and Editor.
The review article titled “Vesicovaginal fistulas: prevalence, impact, and management challenges” which was submitted to Medicina and received the valuable opportunity to be revised by you, has been considered and revised in accordance with your suggestions. Our research team contains two gynecologists who specialize in female pelvic medicine and reconstructive surgery and one urologist. We intend to make an article that can provide data for better urogynecology healthcare and clinical implications in the future. Thank you for your kindly response and very informative comments to improve the quality of our research. Our research team tries so hard to improve our article to make it the best and get the opportunity to publish in your journal. Moreover, we believe that our article is proper, and it will provide interesting data for many readers in the future, which can lead to many citations. We are looking forward to getting the good news of our acceptance to publish in your journal.
We would like to inform the editors and all reviewers that we have amended all of the reviewers' suggestions and expanded the article to exceed 4000 words in accordance with journal standards. Furthermore, we had the institute language service rewrite and proofread the English.
Thank you for the constructive comments, suggestions, and critiques. We have responded point-by-point below in RED and addressed them in the manuscript using track changes.
Yours sincerely,
Associate Professor Orawee Chinthakanan
Apisith Saraluck,MD.
Corresponding author: E-mail: orawee.chi@mahidol.ac.th apisith.sar@mahidol.ac.th
Comments and Suggestions for Authors: Reviewer 1
This review addresses the prevalence, impact, and management challenges of VVF.
The main concern is the lack of work novelty. The authors should emphasize the novelty of this study since there are many review articles on this topic.
Response: Thank you so much for the best comments that could improve the quality of our research.
We add more detail and novelty of treatment and investigation of VVF as your suggestion.

Reviewer 2 Report
Comments and Suggestions for Authors
The paper is of interest, well written and shows a complete review of the literature.
Simply check at page 7 row199: the following sentence: "The urethral catheter is generally removed 10 to 14 days postoperatively." seems to me redundant as the concept has already been presented some sentences above.
Author Response
Dear Reviewer(s) and Editor.
The review article titled “Vesicovaginal fistulas: prevalence, impact, and management challenges” which was submitted to Medicina and received the valuable opportunity to be revised by you, has been considered and revised in accordance with your suggestions. Our research team contains two gynecologists who specialize in female pelvic medicine and reconstructive surgery and one urologist. We intend to make an article that can provide data for better urogynecology healthcare and clinical implications in the future. Thank you for your kindly response and very informative comments to improve the quality of our research. Our research team tries so hard to improve our article to make it the best and get the opportunity to publish in your journal. Moreover, we believe that our article is proper, and it will provide interesting data for many readers in the future, which can lead to many citations. We are looking forward to getting the good news of our acceptance to publish in your journal.
We would like to inform the editors and all reviewers that we have amended all of the reviewers' suggestions and expanded the article to exceed 4000 words in accordance with journal standards. Furthermore, we had the institute language service rewrite and proofread the English.
Thank you for the constructive comments, suggestions, and critiques. We have responded point-by-point below in RED and addressed them in the manuscript using track changes.
Yours sincerely,
Associate Professor Orawee Chinthakanan
Apisith Saraluck,MD.
Corresponding author: E-mail: orawee.chi@mahidol.ac.th apisith.sar@mahidol.ac.th
Comments and Suggestions for Authors: Reviewer 2
The paper is of interest, well written and shows a complete review of the literature.
Response: Thank you so much for your support. We glad that you feel this way to this manuscript.
Simply check at page 7 row199: the following sentence: "The urethral catheter is generally removed 10 to 14 days postoperatively." seems to me redundant as the concept has already been presented some sentences above.
Response: We respect the reviewer's opinion. We revised and deleted the redundant sentence as your recommend.

Reviewer 3 Report
Comments and Suggestions for Authors
It is a good overview regarding the topic VVF. All important aspects were mentioned. Also interesting is the historical background.
I have three comments:
1. "He reported successful VVF repair using silver wire after 30 attempts on 44 his slave in 1952": Fortunately, slavery was abolished before that. Please correct the year.
2. I would like to see a more differentiated presentation of the studies on conservative therapy with regard to catheterization.
3. There are studies on the risk of fistula in hysterectomy depending on the approach. This would be an interesting fact to add.
Author Response
Dear Reviewer(s) and Editor.
The review article titled “Vesicovaginal fistulas: prevalence, impact, and management challenges” which was submitted to Medicina and received the valuable opportunity to be revised by you, has been considered and revised in accordance with your suggestions. Our research team contains two gynecologists who specialize in female pelvic medicine and reconstructive surgery and one urologist. We intend to make an article that can provide data for better urogynecology healthcare and clinical implications in the future. Thank you for your kindly response and very informative comments to improve the quality of our research. Our research team tries so hard to improve our article to make it the best and get the opportunity to publish in your journal. Moreover, we believe that our article is proper, and it will provide interesting data for many readers in the future, which can lead to many citations. We are looking forward to getting the good news of our acceptance to publish in your journal.
We would like to inform the editors and all reviewers that we have amended all of the reviewers' suggestions and expanded the article to exceed 4000 words in accordance with journal standards. Furthermore, we had the institute language service rewrite and proofread the English.
Thank you for the constructive comments, suggestions, and critiques. We have responded point-by-point below in RED and addressed them in the manuscript using track changes.
Yours sincerely,
Associate Professor Orawee Chinthakanan
Apisith Saraluck,MD.
Corresponding author: E-mail: orawee.chi@mahidol.ac.th apisith.sar@mahidol.ac.th
Comments and Suggestions for Authors: Reviewer 3
It is a good overview regarding the topic VVF. All important aspects were mentioned. Also interesting is the historical background.
I have three comments:
- "He reported successful VVF repair using silver wire after 30 attempts on 44 his slave in 1952": Fortunately, slavery was abolished before that. Please correct the year.
Response: We appreciate the reviewer's comments. We have revised the manuscript as all suggestions above. (Correct to 1852)
- I would like to see a more differentiated presentation of the studies on conservative therapy with regard to catheterization.
Response: We appreciate the reviewer's comments. We have revised the manuscript as all suggestions above.
- There are studies on the risk of fistula in hysterectomy depending on the approach. This would be an interesting fact to add.
Response: We appreciate the reviewer's comments. We have revised the manuscript as all suggestions above.

Round 2
Reviewer 1 Report
Comments and Suggestions for Authors
The comment has responded satisfactorily.
Comments on the Quality of English LanguageMinor edit is needed.